# Impact of Negative Feedbacks on De Novo Pyrimidines Biosynthesis in *Escherichia coli*

**DOI:** 10.3390/ijms24054806

**Published:** 2023-03-02

**Authors:** Ilya R. Akberdin, Konstantin N. Kozlov, Fedor V. Kazantsev, Stanislav I. Fadeev, Vitaly A. Likhoshvai, Tamara M. Khlebodarova

**Affiliations:** 1Department of Systems Biology, Institute of Cytology and Genetics SB RAS, 630090 Novosibirsk, Russia; 2Department of Natural Sciences, Novosibirsk State University, 630090 Novosibirsk, Russia; 3Biosoft.Ru, Ltd., 630058 Novosibirsk, Russia; 4Department of Computational Biology, Scientific Center for Information Technologies and Artificial Intelligence, Sirius University of Science and Technology, 354340 Sochi, Russia; 5Higher School for Applied Mathematics and Computational Physics, PhysMech, Peter the Great St. Petersburg Polytechnic University, 195251 St. Petersburg, Russia; 6Kurchatov Genomics Center, Institute of Cytology and Genetics, SB RAS, 630090 Novosibirsk, Russia; 7Institute of Mathematics SB RAS, 630090 Novosibirsk, Russia

**Keywords:** in silico cell, *E. coli*, pyrimidine biosynthesis, negative feedbacks, kinetic modeling, oscillations

## Abstract

Earlier studies aimed at investigating the metabolism of endogenous nucleoside triphosphates in synchronous cultures of *E. coli* cells revealed an auto-oscillatory mode of functioning of the pyrimidine and purine nucleotide biosynthesis system, which the authors associated with the dynamics of cell division. Theoretically, this system has an intrinsic oscillatory potential, since the dynamics of its functioning are controlled through feedback mechanisms. The question of whether the nucleotide biosynthesis system has its own oscillatory circuit is still open. To address this issue, an integral mathematical model of pyrimidine biosynthesis was developed, taking into account all experimentally verified negative feedback in the regulation of enzymatic reactions, the data of which were obtained under in vitro conditions. Analysis of the dynamic modes of the model functioning has shown that in the pyrimidine biosynthesis system, both the steady-state and oscillatory functioning modes can be realized under certain sets of kinetic parameters that fit in the physiological boundaries of the investigated metabolic system. It has been demonstrated that the occurrence of the oscillatory nature of metabolite synthesis depended on the ratio of two parameters: the Hill coefficient, *h*_UMP1_—the nonlinearity of the UMP effect on the activity of carbamoyl-phosphate synthetase, and the parameter *r* characterizing the contribution of the noncompetitive mechanism of UTP inhibition to the regulation of the enzymatic reaction of UMP phosphorylation. Thus, it has been theoretically shown that the *E. coli* pyrimidine biosynthesis system possesses its own oscillatory circuit whose oscillatory potential depends to a significant degree on the mechanism of regulation of UMP kinase activity.

## 1. Introduction

The regulation of processes in biological systems (metabolic, molecular, and genetic) by feedback mechanism plays an important role, allowing a living system, on the one hand, to control intracellular processes and maintain homeostasis, and, on the other hand, to adapt to changing environmental conditions. A theoretical analysis of the dynamics of models describing a number of genetic and metabolic systems regulated by the feedback mechanism showed that such systems often demonstrate complex dynamic modes [1,2,3,4]. Thus, if cellular processes are controlled by the negative feedback mechanism, they create conditions for the formation of periodic dynamics [5,6,7,8,9,10], and the combination of positive-negative coupling and double negatives contributes to the manifestation of more complex modes, including quasiperiodic, chaotic, and even hyperchaotic [11,12,13,14,15]. The characteristics of the dynamic modes also depend not only on the number and type of the regulatory loops, but also on their length, i.e., on the number of intermediate components between the control element and its target [16].

From this point of view, the study of dynamic modes of functioning of key metabolic systems orchestrated by the feedback mechanism, which include the nucleotide biosynthesis system, is of exceptional interest. On the one hand, this system provides such fundamental cellular processes as replication and transcription, and, on the other hand, it is directly involved in central metabolic processes in the cell, since pyrimidine and purine nucleotides are not only direct precursors of nucleic acids, DNA and RNA, but also various coenzymes. The dynamics of such global intracellular systems can impose their influence on the mode of functioning of the entire cell.

Continuous nucleotide synthesis is required to maintain the necessary levels of RNA and DNA concentration in the cell. In prokaryotic organisms, 90% of all nucleotides are synthesized through the de novo pathway. On this basis, in the present study, we developed an integral kinetic model of de novo pyrimidine biosynthesis in the cell of *Escherichia coli*, as one of the most studied organisms in the context of regulatory mechanisms, and analyzed the dynamic modes of its functioning. A feature of this system is the presence of a large number of regulatory loops controlling the synthesis of the final product by a feedback mechanism, the influence of which on the dynamics of nucleotide synthesis has not been practically studied.

The existing experimental data suggest that in the biosynthesis of ribo- and deoxyribopyrimidine nucleotides in *E. coli*, the activity of at least six enzymes is controlled by the biosynthesis products through a feedback mechanism [17,18,19,20,21,22]. Figure 1 shows the scheme of regulation of the metabolic pathway of pyrimidine nucleotide biosynthesis de novo in *E. coli*, including the first nine steps leading to CTP synthesis. It can be seen that five enzymes of this stage of biosynthesis are under the control of eight negative regulatory feedbacks of different lengths, most of which are provided not so much by the final product of CTP as by its precursors—UMP, UDP, and UTP. These data suggest the capability of implementing complex dynamic modes in the pyrimidine biosynthesis system.

In early studies aimed at studying the metabolism of endogenous nucleoside triphosphates in synchronous cultures of *E. coli* cells, an auto-oscillatory mode of the deoxyribo- and ribopyrimidine and purine nucleotides biosynthesis system was found, but the authors associated it with the dynamics of cell division [23]. In higher organisms (larvae of *Danio rerio*, in the liver of mice), rhythmic changes in the de novo nucleotide biosynthesis system were also described, which the authors associated with circadian control of the cell cycle [24,25].

The question of whether the nucleotide biosynthesis system possesses its own oscillatory circuit or implements any other dynamic modes remains open. The theoretical prediction of the capability of the existence of oscillation in the intracellular concentration of carbamoyl phosphate, a key precursor of pyrimidines, was made by A. Goldbeter as early as 1973 [26], but the first mathematical models describing the dynamic state of this system were developed only in 2005 and 2009 [27,28]. These models, taking into account the first nine steps leading to the synthesis of the ribopyrimidines UTP and CTP, provided qualitatively different results. For the model [27], in which three negative feedbacks controlling the first two steps of the process were taken into account, a steady-state mode of functioning was obtained. In the model [28], which took into account the influence of four negative feedback loops controlling the first, second, and ninth steps of synthesis, an oscillatory attractor was identified.

The inconsistency of the previously obtained results [27,28] served as the basis for a detailed analysis of the mechanisms of the coordinated synthesis of pyrimidine nucleotides. The present model takes into account the existing experimental data [17,18,19,20,21,22] on the presence of eight loops of negative regulation of CTP synthesis (see Figure 1). An analysis of the dynamic modes of the model, including the combination of DEEP [29] and BaSAR methods [30], showed that both steady-state and oscillatory modes of functioning in the pyrimidine biosynthesis system are possible under certain sets of kinetic parameters that fit into the physiological boundaries of the studied metabolic system. It was shown that both the complexity of the mechanism of regulation by uridine monophosphate (UMP) of the activity of the first enzyme in the pyrimidine synthesis chain, carbamoyl-phosphate synthetase (regulatory loop 1, Figure 1), and also the allosteric type of the mechanism of UMP kinase activity regulation by uridine triphosphate (UTP) (regulatory loop 6, Figure 1) play a decisive role in the formation of the oscillatory mode of functioning of the pyrimidine biosynthesis model.

## 2. Results and Discussion

### 2.1. Integral Mathematical Model of Pyrimidine Biosynthesis

A mathematical model of pyrimidine biosynthesis is reconstructed from elementary models of enzymatic reactions (see Appendix A). The final model is a system (1) of nine ordinary differential equations that describe the rates of change in the concentrations of compounds synthesized or consumed during nucleotide synthesis. The variables of the model are concentrations of compounds x_1_ (CAP), x_2_ (CAASP), … x_9_ (CTP), which correspond to the designations in Figure 1. The other substances included in the elementary models (ATP, IMP,…) act as external substances whose concentrations are set as parameters and do not change in the process of numerical calculations.
(1)$$\begin{array}{lll} \left\{\begin{array}{l} \frac{dx_{1}}{\mathrm{dt}}{=V}_{1}{\;-\;V}_{2}\\ \frac{dx_{2}}{\mathrm{dt}}{=V}_{2}{\;-\;V}_{3}\\ \frac{dx_{3}}{\mathrm{dt}}{=V}_{3}{\;-\;V}_{4}\\ \frac{dx_{4}}{\mathrm{dt}}{=V}_{4}{\;-\;V}_{5}\\ \frac{dx_{5}}{\mathrm{dt}}{=V}_{5}{\;-\;V}_{6}\\ \frac{dx_{6}}{\mathrm{dt}}{=V}_{6}{\;-\;V}_{7}\\ \frac{dx_{7}}{\mathrm{dt}}{=V}_{7}{\;-\;V}_{8}{\;-\;V}_{15}\\ \frac{dx_{8}}{\mathrm{dt}}{=V}_{8}{\;-\;V}_{9}{\;-\;V}_{10}\\ \frac{dx_{9}}{\mathrm{dt}}{=V}_{9}{\;-\;V}_{12}{\;-\;V}_{11} \end{array}\right. & ,\;where & \begin{array}{c} V_{1}=k_{1f}\times E_{1}\times \frac{\frac{BC}{{K_{m}}_{{}_{\mathrm{BC}}}}\times \frac{GLN}{K_{m_{\mathrm{GLN}}}}\times {\left(\frac{\mathrm{ATP}}{K_{{m_{\mathrm{ATP}}}_{{}_{1}}}}\right)}^{2}}{\left(1+\frac{BC}{{K_{m}}_{{}_{\mathrm{BC}}}}\right)\times \left(1+\frac{GLN}{K_{m_{\mathrm{GLN}}}}\right)\times {\left(1+\frac{\mathrm{ATP}}{K_{{m_{\mathrm{ATP}}}_{{}_{1}}}}\right)}^{2}}\times \frac{\left(1+{\;\;\;\delta \;\;\;}_{\mathrm{orn}}\times {\left(\frac{\mathrm{orn}}{k_{{\mathrm{orn}}_{1}}}\right)}^{\mathrm{horn}}\right)}{\left(1+{\left(\frac{\mathrm{orn}}{k_{{\mathrm{orn}}_{1}}}\right)}^{\mathrm{horn}}\right)}\times \frac{\left(1+{\;\;\;\delta \;\;\;}_{\mathrm{IMP}}\times \left(\frac{\mathrm{IMP}}{k_{{\mathrm{IMP}}_{1}}}\right)\right)}{1\;+\;{\left(\frac{x_{6}}{K_{{\mathrm{UMP}}_{1}}}\right)}^{h_{{\mathrm{UMP}}_{1}}}+\;\left(\frac{\mathrm{IMP}}{K_{{\mathrm{IMP}}_{1}}}\right)\;+\;{\left(\frac{x_{7}}{K_{{\mathrm{UDP}}_{1}}}\right)}^{h_{{\mathrm{UDP}}_{1}}}+\;{\left(\frac{x_{8}}{K_{{\mathrm{UTP}}_{1}}}\right)}^{h_{{\mathrm{UTP}}_{1}}}}\\ V_{1}=k_{1}\times \frac{\left({1+\delta }_{\mathrm{IMP}}\times \left(\frac{\mathrm{IMP}}{k_{{\mathrm{IMP}}_{1}}}\right)\right)}{1+{\left(\frac{x_{6}}{K_{{\mathrm{UMP}}_{1}}}\right)}^{h_{{\mathrm{UMP}}_{1}}}+\left(\frac{\mathrm{IMP}}{K_{{\mathrm{IMP}}_{1}}}\right)+{\left(\frac{x_{7}}{K_{{\mathrm{UDP}}_{1}}}\right)}^{h_{{\mathrm{UDP}}_{1}}}+{\left(\frac{x_{8}}{K_{{\mathrm{UTP}}_{1}}}\right)}^{h_{{\mathrm{UTP}}_{1}}}},k_{1}=0.012\frac{mM}{\sec}\\ V_{2}=E_{2}\times k_{2f}\times \frac{{\left(\frac{x_{1}}{K_{m_{\mathrm{CAP}}}}\right)}^{h_{\mathrm{CAP}}}}{1+{\left(\frac{x_{1}}{K_{m_{\mathrm{CAP}}}}\right)}^{h_{\mathrm{CAP}}}}\times \frac{{\left(\frac{asp}{K_{m_{{\mathrm{asp}}_{2}}}}\right)}^{h_{{\mathrm{asp}}_{21}}}}{1+{\left(\frac{asp}{K_{m_{{\mathrm{asp}}_{2}}}}\right)}^{h_{{\mathrm{asp}}_{22}}}}\times \frac{1+{\;\;\;\delta \;\;\;}_{{\mathrm{CTP}}_{2}}\times \left(\frac{x_{9}}{K_{{\mathrm{CTP}}_{2}}}\right)+{\;\;\;\delta \;\;\;}_{{\mathrm{ATP}}_{2}}\times \left(\frac{\mathrm{ATP}}{K_{{\mathrm{ATP}}_{2}}}\right)+{\;\;\;\delta \;\;\;}_{{\mathrm{UTP}}_{2}}\times \left(\frac{x_{8}}{K_{{\mathrm{UTP}}_{2}}}\right)}{1+\left(\frac{x_{9}}{K_{{\mathrm{CTP}}_{2}}}\right)+\left(\frac{\mathrm{ATP}}{K_{{\mathrm{ATP}}_{2}}}\right)+\left(\frac{x_{8}}{K_{{\mathrm{UTP}}_{2}}}\right)+{\;\;\;\omega \;\;\;}_{2}\times \left(\frac{x_{9}}{K_{{\mathrm{CTP}}_{2}}}\right)\times \left(\frac{x_{8}}{K_{{\mathrm{UTP}}_{2}}}\right)},\\ V_{2}=k_{2}\times \frac{{\left(\frac{x_{1}}{K_{m_{{\mathrm{CAP}}_{2}}}}\right)}^{h_{\mathrm{CAP}}}}{1+{\left(\frac{x_{1}}{K_{m_{{\mathrm{CAP}}_{2}}}}\right)}^{h_{\mathrm{CAP}}}}\times \frac{{1+\delta }_{{\mathrm{CTP}}_{2}}\times \left(\frac{x_{9}}{K_{{\mathrm{CTP}}_{2}}}\right){+\delta }_{{\mathrm{ATP}}_{2}}\times \left(\frac{\mathrm{ATP}}{K_{{\mathrm{ATP}}_{2}}}\right){+\delta }_{{\mathrm{UTP}}_{2}}\times \left(\frac{x_{8}}{K_{{\mathrm{UTP}}_{2}}}\right)}{1+\left(\frac{x_{9}}{K_{{\mathrm{CTP}}_{2}}}\right)+\left(\frac{\mathrm{ATP}}{K_{{\mathrm{ATP}}_{2}}}\right)+\left(\frac{x_{8}}{K_{{\mathrm{UTP}}_{2}}}\right){+\omega }_{2}\times \left(\frac{x_{9}}{K_{{\mathrm{CTP}}_{2}}}\right)\times \left(\frac{x_{8}}{K_{{\mathrm{UTP}}_{2}}}\right)},k_{2}=0.31\frac{mM}{\sec}\\ \begin{array}{c} V_{3}=\frac{E_{3}\times k_{3f}\times \frac{x_{2}}{K_{m_{{\mathrm{CAASP}}_{3}}}}}{1+\frac{x_{2}}{K_{m_{{\mathrm{CAASP}}_{3}}}}},\;V_{3}=k_{3}\times \frac{\frac{x_{2}}{K_{m_{{\mathrm{CAASP}}_{3}}}}}{1+\frac{x_{2}}{K_{m_{{\mathrm{CAASP}}_{3}}}}},k_{3}=0.351\frac{mM}{\sec} \end{array}\\ V_{4}=\frac{E_{4}\times \left(k_{4f}\times \frac{Q}{K_{m_{Q}}}\times \frac{x_{3}}{K_{m_{{\mathrm{DOROA}}_{4}}}}-k_{4r}\times \frac{QH2}{K_{m_{\mathrm{QH}2_{4}}}}\times \frac{x_{4}}{K_{m_{{\mathrm{OROA}}_{4}}}}\right)}{\left(1\;+\;\frac{Q}{K_{m_{Q}}}+\frac{QH2}{K_{m_{\mathrm{QH}2_{4}}}}\right)\times \left(1+\frac{x_{3}}{K_{m_{{\mathrm{DOROA}}_{4}}}}\times \left(1+\frac{x_{4}}{K_{{\mathrm{OROA}}_{4}}}\right)+\frac{x_{4}}{K_{m_{{\mathrm{OROA}}_{4}}}}\right)},V_{4}=\frac{\left(k_{4f}\times \frac{x_{3}}{K_{m_{{\mathrm{DOROA}}_{4}}}}-k_{4r}\times \frac{x_{4}}{K_{m_{{\mathrm{OROA}}_{4}}}}\right)}{k_{4}\times \left(1+\frac{x_{3}}{K_{m_{{\mathrm{DOROA}}_{4}}}}\times \left(1+\frac{x_{4}}{K_{{\mathrm{OROA}}_{4}}}\right)+\frac{x_{4}}{K_{m_{{\mathrm{OROA}}_{4}}}}\right)},\\ k_{4f}=1.27\frac{mM}{\sec},k_{4r}=0.225\frac{mM}{\sec},k_{4}=12.28,\\ V_{5}=\frac{E_{5}\times k_{5f}\times \frac{prpp}{K_{m_{{\mathrm{prpp}}_{5}}}}\times \frac{x_{4}}{K_{m_{{\mathrm{OROA}}_{5}}}}}{1\;+\;\frac{prpp}{K_{m_{{\mathrm{prpp}}_{5}}}}+\frac{x_{4}}{K_{m_{{\mathrm{OROA}}_{5}}}}},\;V_{5}=\frac{k_{5}\times \frac{x_{4}}{K_{m_{{\mathrm{OROA}}_{5}}}}}{k_{5prpp}+\frac{x_{4}}{K_{m_{{\mathrm{OROA}}_{5}}}}},k_{5}=3.9\frac{mM}{\sec},k_{5prpp}=7.5\\ V_{6}=\frac{E_{6}\times k_{6f}\times \frac{x_{5}}{K_{m_{{\mathrm{OMP}}_{6}}}}}{1+\frac{x_{5}}{K_{m_{{\mathrm{OMP}}_{6}}}}},V_{6}=\frac{k_{6}\times \frac{x_{5}}{K_{m_{{\mathrm{OMP}}_{6}}}}}{1+\frac{x_{5}}{K_{m_{{\mathrm{OMP}}_{6}}}}},k_{6}=0.73\frac{mM}{\sec},\\ V_{7}=\frac{E_{7}\times k_{7f}\times x_{6}\times \mathrm{ATP}}{\left(K_{{m_{\mathrm{UMP}}}_{{}_{7}}}\times \left(1+\left(1-r\right)\times {\left(\frac{x_{8}}{K_{{\mathrm{UTP}}_{72}}}\right)}^{h_{{\mathrm{UTP}}_{7}}}\right)+x_{6}\right)\times \left(K_{m_{{\mathrm{ATP}}_{7}}}+\mathrm{ATP}\right)}\cdot \frac{1}{1+r\times {\left(\frac{x_{8}}{K_{{\mathrm{UTP}}_{71}}}\right)}^{h_{{\mathrm{UTP}}_{7}}}}\times f_{7},f_{7}=\frac{1+{\;\;\;\delta \;\;\;}_{{\mathrm{GTP}}_{7}}\;\;\;\times \;\;\;{\left(\frac{\mathrm{GTP}}{K_{{\mathrm{GTP}}_{7}}}\right)}^{h_{{\mathrm{GTP}}_{7}}}}{1+{\left(\frac{\mathrm{GTP}}{K_{\mathrm{GTP}7}}\right)}^{h_{{\mathrm{GTP}}_{7}}}},\\ V_{7}=\frac{k_{7}\times x_{6}}{\left(K_{{m_{\mathrm{UMP}}}_{{}_{7}}}\times \left(1+\left(1-r\right)\times {\left(\frac{x_{8}}{K_{{\mathrm{UTP}}_{72}}}\right)}^{h_{{\mathrm{UTP}}_{7}}}\right)+x_{6}\right)}\cdot \frac{1}{1+r\times {\left(\frac{x_{8}}{K_{{\mathrm{UTP}}_{71}}}\right)}^{h_{{\mathrm{UTP}}_{7}}}},k_{7}=0.393\frac{mM}{\sec}\\ \;V_{8}=\frac{E_{8}\times k_{8f}\times x_{7}\times \mathrm{ATP}}{\left(K_{m_{{\mathrm{UDP}}_{8}}}+x_{7}\right)\times \left(K_{m_{{\mathrm{ATP}}_{8}}}+\mathrm{ATP}\right)},\;V_{8}=\frac{k_{8}\times x_{7}}{K_{m_{{\mathrm{UDP}}_{8}}}+x_{7}},k_{8}=0.11\frac{mM}{\sec}\\ V_{9}=\frac{E_{9}\times k_{9f}\times {\left(x_{8}\right)}^{h_{{\mathrm{UTP}}_{9}}}\times {\left(\mathrm{ATP}\right)}^{h_{{\mathrm{ATP}}_{9}}}}{\left(\left({\left(K_{m_{{\mathrm{UTP}}_{9}}}\right)}^{h_{{\mathrm{UTP}}_{9}}}\right)\times \left(1+\frac{x_{9}}{K_{{\mathrm{CTP}}_{9}}}\right)+{\left(x_{8}\right)}^{h_{{\mathrm{UTP}}_{9}}}\right)\times \left({\left(K_{m_{{\mathrm{ATP}}_{9}}}\right)}^{h_{{\mathrm{ATP}}_{9}}}+\left(1+\frac{\mathrm{ATP}}{K_{{\mathrm{ATP}}_{9}}}\right)\times {\left(\mathrm{ATP}\right)}^{h_{{\mathrm{ATP}}_{9}}}\right)}\times f_{9},f_{9}=\frac{k_{0_{9}}+{\;\;\;\delta \;\;\;}_{{\mathrm{GTP}}_{9}}\;\;\;\times \;\;\;\left(\frac{\mathrm{GTP}}{K_{{\mathrm{GTP}}_{91}}}\right)+{\;\;\;\delta \;\;\;}_{\mathrm{GT}P_{91}}\;\;\;\times \;\;\;{\left(\frac{\mathrm{GTP}}{K_{\mathrm{GT}P_{92}}}\right)}^{h_{{\mathrm{GTP}}_{9}}}}{1+\left(\frac{\mathrm{GTP}}{K_{{\mathrm{GTP}}_{91}}}\right)+{\left(\frac{\mathrm{GTP}}{K_{\mathrm{GT}P_{92}}}\right)}^{h_{{\mathrm{GTP}}_{9}}}},\;\\ V_{9}=\frac{k_{9}\times {x_{8}}^{h_{{\mathrm{UTP}}_{9}}}}{\left(\left({\left(K_{m_{{\mathrm{UTP}}_{9}}}\right)}^{h_{{\mathrm{UTP}}_{9}}}\right)\times \left(1+\frac{x_{9}}{K_{{\mathrm{CTP}}_{9}}}\right)+{\left(x_{8}\right)}^{h_{{\mathrm{UTP}}_{9}}}\right)},k_{9}=1.74\times 10^{-5}\frac{mM}{\sec}\\ V_{10}=k_{10}\times x_{8},\;V_{11}=k_{11}\times x_{9},\;V_{12}=k_{23}\times x_{9},\;V_{15}=k_{22}\times x_{7} \end{array} \end{array}$$

### 2.2. Adaptation of Model (1) to Experimental Data

The adaptation of mathematical models of some elementary subsystems of pyrimidine biosynthesis to known experimental data are given in Appendix A. When building the model (1), nine new parameters $k_{1}$, … ,$\;k_{9}$ were calculated based on the parameters of elementary models of enzymatic reactions, which are shown in Appendix A. In sense $k_{i}$ (i = 1, 2, … , 9), parameters reflect the generalized efficiency of enzyme functioning and are equal to the products of concentration of the corresponding enzyme by its catalytic constant, taking into account the effect of external substances, whose concentrations are given as parameters (e.g., BC, GLN, ATP), while *k*_10_, *k*_11_, *k*_22_, *k*_23_ are the rate constants of utilization of the corresponding substances and are responsible for the intensity of consumption of the corresponding metabolites in DNA and RNA synthesis. Models of all enzymatic reactions are manually annotated and loaded into the MAMMOTh database (https://mammoth.sysbio.cytogen.ru/ (accessed on 27 February 2023)) [31], which allows them to be used for automatic reconstruction of complex models, including other *E. coli* biomolecular systems, based on the modular principle of construction [32,33].

Unfortunately, we found no experimental data on changes in the concentration of metabolites formed during the biosynthesis of pyrimidine nucleotides in an *E. coli* cell, and, therefore, to adapt the model and to determine unmeasured parameter values, we used the steady-state concentrations of some metabolites (carbamoyl aspartate, dihydroorotate, UMP, UDP, UTP, CTP) from two independent measurements [34,35], which were obtained at the same dilution rate of 0.1 h^−1^ but between which significant quantitative differences were observed for most metabolites (see Appendix A). The reason for the parametric differences may be related to the molecular-genetic and physiological features of the *E. coli* K12 strains NCM3722 and BW25113, which were used in [34,35], respectively. For example, the ATP and IMP concentrations for the strain BW25113 were 0.33 and 0.098 mM, respectively, whereas for the NCM3722 strain, these values were significantly higher, 9.6 and 0.27 mM. Parameter fitting of the model to these sets of experimental data [34,35] was performed using the DEEP method [29] and the test [36]. To overcome statistically significant differences (*p* < 0.05) between these parameter sets, normalization coefficients were used (Appendix A).

Ultimately, two sets of parameters were determined, one of which was manually adapted to the experimental data [34] (Appendix A, column “manual”, Appendix A), and the other, obtained using the DEEP method [29], allowed the best simultaneous reproduction of different groups of experimental data for *E. coli* (Appendix A, column “best”, Appendix A).

### 2.3. Analysis of the Dynamic Modes of Functioning of the Model (1) De Novo Pyrimidine Nucleotide Synthesis

Numerical calculations of the model (1) adapted to the experimental data [34] (Appendix A, column “manual,” Appendix A) demonstrate that, with this set of parameters, the steady-state mode in metabolite concentrations of the pyrimidine biosynthesis system is realized in the model (see Figure 2). It can be seen that the system contains a significant amount of only two metabolites, UMP and UTP (see Figure 2a), which are negative regulators of this system. According to the scheme shown in Figure 1, loops 1, 3, 5, and 6 of negative regulation are realized in the system that leads to a low level of synthesis of the final CTP product (see Figure 2b).

A numerical study of the model solutions (1) was carried out using the sensitivity analysis built into the BioUML platform [37] and the parameter continuation method [38,39] (Appendix A). Numerical analysis of the model allowed us to reveal the minimum set of parameters, changes in the values of which most strongly affect the model functioning mode and lead to oscillations in the metabolic system of pyrimidine biosynthesis. Among them are the coefficients reflecting the generalized efficiency of the functioning of the enzymes: aspartate transcarbamylase (*k*_2_), dihydroorotase (*k*_3_), and CTP synthetase (*k*_9_), as well as the nonlinearity coefficient of UMP effect on the activity of carbamoyl-phosphate synthetase. The values of the parameters are given in Table 1. It is worth noting that the numerical analysis of the model also showed the most significant role of the parameter *r*, which characterizes the contribution of the noncompetitive mechanism of UTP inhibition in the regulation of the enzymatic reaction of UMP phosphorylation to the numerical solution of the model.

The dynamics of the synthesis of intermediate and final products of the pyrimidine system for a given set of parameters is shown in Figure 3. It can be seen that a significant amount of five metabolites is observed in the system: CAP, CASP, UMP, UTP and CTP, three of which, namely, UMP, UTP and CTP, are negative regulators of this system. It can also be seen that UMP is the key factor in the regulation of the dynamic mode of the pyrimidine system, whose oscillating dynamics of synthesis seems to shape the synthesis mode of carbamoyl phosphate and carbamoyl aspartate through regulatory loop 1 (see Figure 1), although the contribution of UTP through regulatory loops 3, 5, and 6 is not excluded.

Apparently, UTP also plays a key role in the regulation of the level of pyrimidine synthesis, a significant amount of which can affect the activity of at least three enzymes of the pyrimidine synthesis chain: carbamoyl-phosphate synthetase, aspartate transcarbamoylase, and UMP kinase, via regulatory loops 3, 5, and 6. The contribution of CTP to the regulation of the system is less significant and is realized through the 4th and 7th negative loops. In this system, negative loops 2 and 8 are not implemented due to the lack of accumulation of significant amounts of orotate and UDP.

It should be noted that the values of the varying parameters affecting the occurrence of oscillatory dynamics in model (1), namely, the Hill parameter ($h_{{\mathrm{UMP}}_{1}}$) for the first reaction of pyrimidine nucleotide biosynthesis catalyzed by carbamoyl-phosphate synthetase, and the generalized constant of the enzymatic reaction efficiency, *k*_9_, catalyzed by CTP synthetase, lie within the range of values obtained by adapting model (1) to various sets of experimental data (see Table 1). At the same time, the values of generalized constants, *k*_2_ and *k*_3_, of the efficiency of enzymatic reactions catalyzed by aspartate transcarbamylase and dihydroorotase, respectively, are significantly lower. Are these values physiologically significant?

As can be seen from model (1), the values of these parameters are significantly determined by the concentration of the enzyme in the cell, which in *E. coli* can differ by almost two orders of magnitude [40,41], which is especially characteristic for aspartate transcarbamylase. The value of the constant *k*_2_ also depends on *K_m_*, *k_cat_* of the enzyme, and the *h_asp_* coefficient. In studies devoted to the analysis of the kinetic properties of aspartate transcarbamylase in *E. coli*, these parameters also vary. Thus, *K_m_* (*ASP*_0.5_) values vary from 6 to 39 mM [17,42,43], *k_cat_* from 240 to 1667 s^−1^ [44,45,46], while *h_asp_* ranges from 1.6 to 4.19 [42,47,48,49], and in other bacterial species *h_asp_* > 10 under certain conditions [50]. That is, even at the maximum known concentration of aspartate transcarbamylase in an *E. coli* cell of ~0.008 mM [41], the value of *k*_2_ at which oscillations are observed in model (1) is more than an order of magnitude greater than the minimum feasible value.

As for the parameter *k*_3_, its value is determined only by the level of dihydroorotase in the cell, which in *E. coli* varies from ~0.00134 to 0.0041 mM [41], and by the parameter *k_cat_*. In this case, the value of *k_3_
*= 0.00351 mM/sec at which oscillations occur may only be feasible for an enzyme with low catalytic activity, which is typical for some bacterial species (*k_cat_* = 1.4–2.48 s^−1^) [51,52,53] and protists (*k_cat_* = 0.073–2.1 s^−1^) [54,55,56].

At the same time, it should be noted that in *E. coli* the *k_cat_* values for dihydroorotase (PyrC) measured in vitro under “optimal conditions” range from 143 to 195 s^−1^ [57,58], which is almost two orders of magnitude higher than the values estimated for this enzyme from in vivo omics data [59]. The possibility of a significant discrepancy between in vitro and in vivo enzyme kinetic properties is also evidenced by other data [60,61,62].

Thus, the possibility that the values of model parameters (1) leading to the oscillations may lie within the physiological limits of pyrimidine system functioning and be the cause of the oscillations of metabolites in the *E. coli* cell in vivo is not ruled out. Moreover, the leading role in the occurrence of oscillatory dynamics of metabolite synthesis in the pyrimidine synthesis system de novo belongs to UMP and possibly UTP, which regulate the activity of carbamoyl-phosphate synthetase, aspartate transcarbamoylase, and UMP kinase by the feedback mechanism through the interaction of at least three negative loops.

For the model, the adaptation of which was carried out by the DEEP method simultaneously to two sets of experimental data (see Appendix A, column “best,” Appendix A), similar results were obtained, which, however, also indicated that the occurrence of an oscillatory nature of the metabolite synthesis in the pyrimidine system depends on the ratio of two parameters: the Hill coefficient $h_{{\mathrm{UMP}}_{1}}—$the nonlinearity of the effect of UMP on the activity of carbamoyl-phosphate synthetase (regulatory loop 1) and the parameter *r*, which characterizes the contribution of the noncompetitive mechanism of UTP inhibition to the regulation of the enzymatic reaction of UMP phosphorylation (regulatory loop 6).

Figure 4 shows the characteristic oscillatory profile for the synthesis of metabolites of the pyrimidine system at the values of $h_{{\mathrm{UMP}}_{1}}$ = 6 and *r* = 0.28. It should be noted that the value of $h_{{\mathrm{UMP}}_{1}}$ = 6, is quite high in terms of the complexity of the enzymatic reaction catalyzed by carbamoyl-phosphate synthetase, while the value of *r* = 0.28 indicates the possible contribution of a competitive mechanism of UTP inhibition to the regulation of UMP kinase activity.

In the next section, the effect of the complexity of the negative feedbacks on the dynamics of pyrimidine metabolism functioning in the *E. coli* cell was investigated in terms of two hypotheses on the mechanism of UTP inhibition of UMP kinase activity—competitive [63] and noncompetitive [64], whose effect on the dynamics of pyrimidine synthesis was noted above (parameter *r*).

### 2.4. Effect of the Complexity of Negative Feedbacks on the Functioning Mode of the Model (1)

In terms of the effect of the complexity of the regulatory loops on the dynamics of the pyrimidine synthesis system, we analyzed the effect of $h_{{\mathrm{UMP}}_{1}}$,$\;h_{{\mathrm{UDP}}_{1}}$, and$\;h_{{\mathrm{UTP}}_{1}}$ parameter values describing the nonlinearity of the inhibitory effects of the respective biosynthesis products on the first enzymatic reaction of this metabolic pathway catalyzed by carbamoyl-phosphate synthetase.

The theoretical study was carried out using the BaSAR method [30] for different variants of the basic model: (a) the model adapted by the manual method to the data of [34] (Appendix A, column “manual”, Appendix A); (b) the model in which the oscillatory mode of operation was obtained based on the manual method (values of parameters *h_ump_*_1_, *k*_2_, *k*_3_, and *k*_9_ from Table 1, column “oscillation”); and (c) the model adapted by the DEEP method to two independent sets of experimental data in which the oscillatory mode of operation was obtained (Appendix A, column “best”, Appendix A).

As a result, it was shown that for model “a”, no oscillatory mode of operation is observed for any combination of the Hill parameters in the range of their variation [1;10] and the parameter *r* ∈ [0;1] (Figure 5a; Appendix A), while for models “b” and “c” there are regions of oscillations at the corresponding parameter sets (Figure 5b,c; Appendix A).

Ultimately, the analysis of models “b” and “c” demonstrated that with the noncompetitive mechanism (*r* = 1) of UTP inhibition of the enzymatic reaction of uridine monophosphate phosphorylation (regulatory loop 6), the probability of oscillatory behavior in the system under study increases dramatically compared to the competitive mechanism. Moreover, the existence of an oscillatory mode for the mixed mechanism of this enzymatic reaction and the values of the Hill parameters in the ranges of variation that are physiologically justified for the first enzymatic reaction of pyrimidine biosynthesis was shown for model “c”.

## 3. Materials and Methods

### 3.1. Methods of the Numerical Analysis

We numerically solved the model equations by using ode15s solver in MATLAB version of the model and an ordinary differential equation JVode solver built into the BioUML for SBML version of the model [37], with default settings and time increment equals to 30 s. Sensitivity analysis implemented in the BioUML was utilized to investigate the effect of the parameter change on the model solution. The search for periodic solutions of the model was performed using the algorithms of the STEP package [38,39]. A description of the methods is given in Appendix A.

The dynamic modes of the model were analyzed using a combination of DEEP [29] and Bayesian Spectrum Analysis (BaSAR) [30] methods.

### 3.2. Methods of the Parameter Fitting

When adapting the model of pyrimidine nucleotide biosynthesis, we used the manual method of parameter selection, since the values of almost all model parameters were known as well as the software implementation of the DEEP method [29], which is a modification of the global stochastic approach and the optimal shortest descent method for finding the minimum functionality (http://sourceforge.net/projects/deepmethod/ (accessed on 27 February 2023)). The method for adaptation of the model to the available data on the steady-state concentration of the metabolites of the system under study is given in Appendix A.

## 4. Conclusions

In *E. coli* metabolic systems, periodic changes in the concentration of metabolites play a crucial role in the regulation of cell division, cell response to the toxic effects of nitric oxide, etc. [65,66,67]. It is known that conditions for the formation of periodic dynamics arise in the presence of negative feedback mechanisms of regulation of cellular processes in the system [5,6,7,8,9,10]. Such mechanisms are characteristic of the pyrimidine nucleotide biosynthesis system in *E. coli* [17,18,19,20,21,22]. The opportunity of oscillations in the concentration of the key pyrimidine precursor, carbamoyl phosphate, was already theoretically predicted in 1973 [26]; however, the general question about the presence of the nucleotide biosynthesis system’s own oscillatory circuit has remained open.

In our study, we performed a theoretical analysis of a mathematical model of pyrimidine biosynthesis that takes into account all experimentally verified negative feedbacks of the regulation of enzymatic reactions in this metabolic pathway. The simulation results suggest the presence of oscillatory potential in the system of pyrimidine de novo synthesis in *E. coli*, but its implementation to a significant extent depends on the mechanisms of UMP kinase activity regulation via UTP—competitive [63] or noncompetitive [64]. Moreover, the possibility of an oscillatory mode in the pyrimidine synthesis system under the mixed mechanism of regulation of this enzymatic reaction has been shown.

It should be noted here that the simulation results are based on the kinetic properties of enzymes that were established under in vitro conditions. Currently, data are accumulating on a significant discrepancy between the kinetic properties of enzymes in vitro and in vivo [59,60,61,62]. However, we believe that these differences do not invalidate the intrinsic catalytic mechanisms of the occurrence of certain enzymatic reactions and, under certain conditions, require their clarification, especially if they are key in any processes. In the system under study, this concerns a more detailed study of the mechanism of the UMP phosphorylation reaction, possibly at the level of a single cell in vivo.

From our point of view, the very fact that the pyrimidine synthesis system may have its own oscillatory circuit provided by the interaction of the negative feedbacks controlling the synthesis of intermediate products is important. Fluctuations of these intermediates concentration under certain conditions will certainly affect the synthesis of DNA and RNA, i.e., the functioning dynamics of such global intracellular systems as replication and transcription, and possibly the functioning of the entire cell.

## Figures and Tables

**Figure 1 ijms-24-04806-f001:**
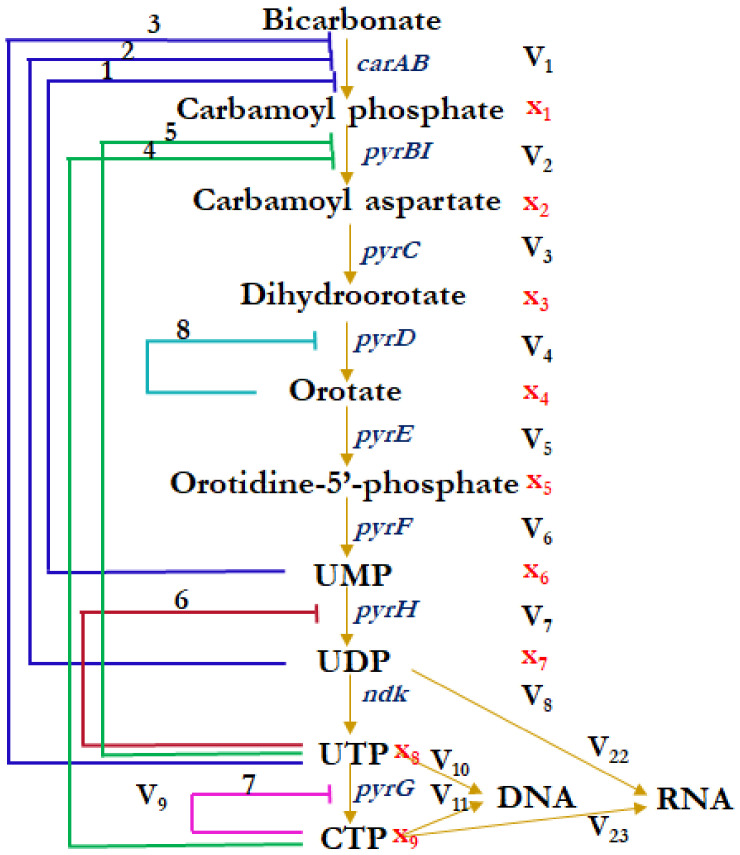
A scheme of the regulation of the metabolic pathway of de novo pyrimidine nucleotide biosynthesis in *E. coli*. Arrows represent the direction of enzymatic reactions; lines ending at the blunt end indicate loops of negative regulation of enzyme activity by biosynthetic products. The blue color of the regulatory loops reflects the regulation of the first enzymatic reaction controlled by carbamoyl-phosphate synthetase (EC 6.3.5.5); green represents the second, aspartate transcarbamoylase (EC 2. 1.3.2); sea wave—fourth, dihydroorotate dehydrogenase (EC 1.3.5.2); red—seventh, UMP kinase (EC 2.7.4.14, 2.7.4.22); and pink—ninth, CTP synthetase (EC 6.3.4.2). The numbers above the negative feedbacks denote the sequence number of the regulatory relationship in the model analysis—the considered stages of pyrimidine nucleotide biosynthesis are controlled by 8 regulatory loops. V_j_ (j = 1,…,9)—rates of enzymatic reactions; V_10_, V_11_, V_22_, V_23_—outflow rates; X_j_ (j = 1,…,9)—model variables—concentrations of substances. Full names are given for the first six substances, and generally accepted abbreviations are given for the others.

**Figure 2 ijms-24-04806-f002:**
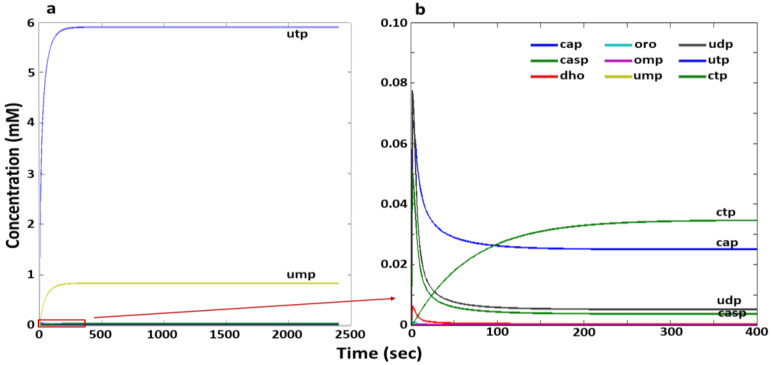
Simulation results of the pyrimidine biosynthesis model in the *E. coli* cell adapted manually to the experimental data (Appendix A, column “manual,” Appendix A). The *X* axis is time (sec), the *Y* axis is the concentration of metabolites (mM). Designations: cap—carbamoyl phosphate, casp—carbamoyl aspartate, dho—dihydroorotate, oro—orotate, omp—orotidine monophosphate, ump—uridine monophosphate, udp—uridine diphosphate, utp—uridine triphosphate, ctp—cytidine triphosphate. The red rectangle in (**a**) marks the area that corresponds to (**b**). The color labels of the metabolite concentration curves in Figures (**a**,**b**) are the same.

**Figure 3 ijms-24-04806-f003:**
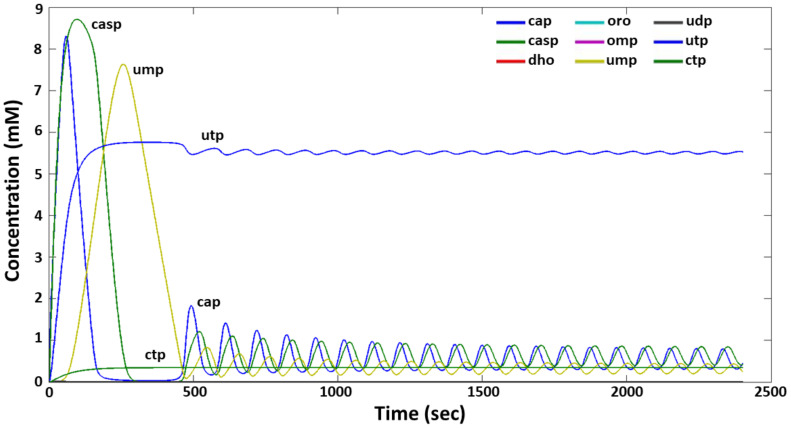
The oscillatory mode of functioning with manual method of the model analysis at *h*_UMP1_ = 2.1, *k*_2_ = 0.0031, *k*_3_ = 0.00351, *k*_9_ = 0.000174 and *r* = 1. Values of other parameters are given in Appendix A (column “manual”, Appendix A). The *X* axis is time (sec), the *Y* axis is the concentration of metabolites (mM). The designations of metabolites and colors of their concentration curves are similar to those in Figure 2.

**Figure 4 ijms-24-04806-f004:**
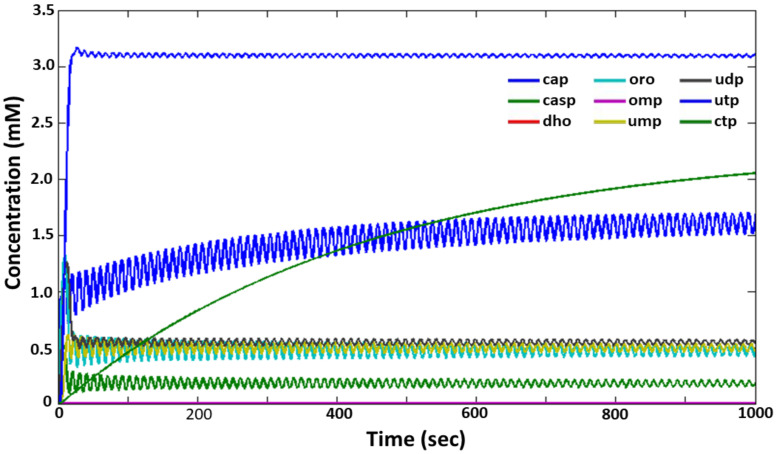
The oscillatory mode of functioning of the pyrimidine synthesis system in the DEEP model analysis (*h*_UMP1_ = 6 and *r* = 0.28). Values of other parameters are given in Appendix A (column “best”, Appendix A). The *X* axis is time (sec), the *Y* axis is the concentration of metabolites (mM). The designations of metabolites and colors of their concentration curves are similar to those in Figure 2.

**Figure 5 ijms-24-04806-f005:**
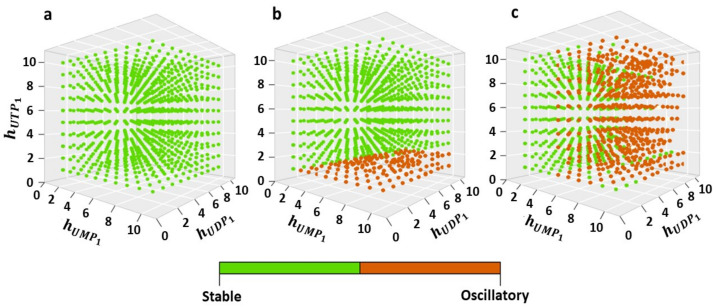
The dynamic state of the pyrimidine biosynthesis metabolic system determined using the Bayesian Spectrum Analysis method [30; Appendix A] in the three-dimensional space of Hill parameters (*h*_UMP1_, *h*_UDP1_, *h*_UTP1_) at parameter value *r* = 1, which corresponds to noncompetitive UTP inhibition of the first pyrimidine nucleotide phosphorylation enzymatic reaction, UMP. The green dots correspond to the stationary state of the system, and the orange dots correspond to the oscillatory mode of functioning. Each of the Hill parameters was varied in increments of one. The full version of the analysis of the three model variants with variation of *r* = [0;1] is given in Appendix A.

**Table 1 ijms-24-04806-t001:** Parameters of the model (1), which affect the occurrence of oscillations in concentrations of metabolites in the pyrimidine biosynthesis system, and their values.

Parameter	Value of the Parameter		
	Appendix A, “Manual”	Appendix A, “Best”	Oscillation
*h_ump_* _1_	1.4	4	2.1
*k*_2_, mM/s	0.31	0.073	0.0031
*k*_3_, mM/s	0.351	0.486	0.00351
*k*_9_, mM/s	0.0000174	0.000174	0.000174

## Data Availability

The integrated kinetic model of pyrimidine biosynthesis is available in MATLAB and SBML formats through the web interface of BioUML software at gitlab project: https://gitlab.sirius-web.org/ecoli_models/ecoli_pyr_model (accessed on 27 February 2023) and mechanistic models for each enzymatic reaction can be found at: https://mammoth.sysbio.cytogen.ru/api (accessed on 27 February 2023).

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
