# Peer review of "Impact of Negative Feedbacks on De Novo Pyrimidines Biosynthesis in *Escherichia coli"

_ijms, 2023, doi:10.3390/ijms24054806_

Round 1

Reviewer 1 Report

The authors present a computer model that reconstructs the emergence of oscillations from the feedback loops involved in the regulation of pyrimidine nucleotide biosynthesis in E.coli.

The study is well-designed and well-presented, with extensive support from the literature.  The techniques are well described. The assumptions and conclusions are well justified. To my opinion, the manuscript is ready for publication in its current form.

Perhaps, I could suggest several minor improvements:

Figure 1. The yellow color of reaction names (V1, V2, etc) and enzymes (pyrF, etc) is not very visible, especially in the paper-printed version (depends a lot on the quality of the printer). Why not take a more contrasting color, e.g. black or dark blue?

In the model description (page 4), on the right-hand side from systems of ODEs, just above v1=k1*, there is symbol «,где» was misplaced from the equation above.

Page 10. There is a dot  “.” in the space between paragraphs 4 and 5.

Author Response

The file with reply to reviewers is attached.

Reviewer 2 Report

In this manuscript, Akberdin et. al. proposed a model of pyrimidine biosynthesis, analyzed  a few scenarios, and discussed the potential oscillatory circuit. Overall, I found this manuscript well-written, the analysis part is comprehensive, which uses both real experimental data and model fitting data, and the discussion part covers both the advantages and disadvantages of this model. Even if there may be more regulation of the pathway discovered, or the in vivo data might be different, I think this manuscript still provides a good way of thinking about how to model such problems.

Minor issues:

  1. The font of the formula has some slight differences.
  2. I suggest the authors to provide some important experiment data from the supplemental text to the main text for better understanding

Author Response

(The authors gave the same response as above.)
